# The Impact of Compressing MBA Courses on Student Satisfaction: Empirical Results

William Swart *[ID] and Christine M. Kowalczyk

Thomas G. Arthur Graduate School of Business, East Carolina University, Greenville, NC 27858, USA;
kowalczykc@ecu.edu
* Correspondence: swartw@ecu.edu

**Abstract:** A Master of Business Administration (MBA) degree is the most popular graduate degree available. It is valuable to those who work in a wide range of business management areas or to prospective entrepreneurs. Many universities have developed compressed 8-week semesters to better meet student needs. Critics question instructor course delivery and student learning. We examine the impact of compressed semesters on MBA student satisfaction and learning in both face-to-face and online courses. Five instructors were assigned courses to be taught in a 16-week semester, followed by the same courses in an 8-week semester. At least one section of the courses was taught in face-to-face and/or online formats. Student satisfaction was measured via transactional distances as well as by their willingness to recommend the course to a friend. A total of 602 responses were obtained, of which 402 were usable. Stepwise multiple regression was used to measure satisfaction. Logistic regression was used to determine which factors influenced students to recommend the course to a friend. The results indicated that there was no reason to believe that there was a significant difference in student satisfaction, nor in their willingness to recommend the course. We also did not find a difference in learning outcomes.

**Keywords:** student satisfaction; student engagement; compressed semester; transactional distance; stepwise regression; logistic regression; learning outcomes

## 1. Introduction

A Master of Business Administration (MBA) degree is the most popular graduate degree in the United States, with 202,334 earned in the 2020–2021 academic year [1]. An MBA degree focuses on acquiring an understanding of business principles while developing leadership skills. It is valuable to those individuals who are interested in working in a wide range of business management areas or to prospective entrepreneurs. According to the *Harvard Business Review*, individuals seeking an MBA will gain a larger professional network, career opportunities, and a bigger salary; however, there are costs including time, tuition, and effort [2].

While online education increased during the COVID-19 pandemic, the Graduate Management Admission Council (GMAC) has reported that the enrollment in *Fortune*'s top 100 online MBA programs decreased by 12.5% between the 2021–2022 and 2022–2023 academic years; however, other programs experienced up to 50% growth in enrollment [3]. With a reported 5% decrease in overall applications to MBA programs around the world, flexible programs—online, part-time, and hybrid—saw an increase in the number of applications, signifying a shift in preferences of MBA students [4]. The Association to Advance Collegiate Business Schools (AACSB) has reported that for online MBA programs to maintain competitiveness, they need to offer flexible formats, including hybrid, accelerated, and customizable skills-based programs, to meet the changing needs and demands of students [5].

The purpose of this study is to evaluate whether the compressed 8-week semester at a large university in the southeastern United States better meets student needs compared to

the full-term (16 week) semesters. In addition, the impact of the compressed semester on MBA student engagement and satisfaction with learning in both face-to-face and online courses is analyzed. This university has an MBA enrollment of 618 online students and 96 face-to-face students. It should be noted, as explained later in the paper, that compressing face-to-face courses may require them to be converted to hybrid courses. Thus, this paper examines compressed courses that have both online and face-to-face components.

## 2. Background Information

More than 555 universities in the United States offer MBA degrees, with another 1062 universities offering MBA courses abroad [6]. Thus, there is fierce competition among universities to enroll and maintain students. To maintain students and grow MBA programs, universities are compressing regular 16-week semester-long courses into shorter mini-semesters lasting 8 weeks or less. While Lutes and Davis [7] conducted a study at Brigham Young University involving compressed face-to-face undergraduate general education courses, we have not found similar studies for graduate online courses. Furthermore, little research has evaluated the impact of this competitive strategy on student engagement and satisfaction, including face-to-face and online formats, specifically in MBA programs.

According to Assistant Dean of Career Services Liza Kirkpatrick at the Kellogg School of Management at Northwestern University, there are many reasons for an individual to consider pursing an MBA, including preparing for a future career, exploring a new industry, accelerating career opportunities, and developing and expanding one's network [1]. However, MBA programs must cater to students who may not have a business undergraduate degree, thus supporting a variety of degrees from any number of fields. Some students may enroll directly after completing their bachelor's degree, while others may be at different career stages, balancing their professional and personal life responsibilities with the academic requirements. MBA programs are challenged to meet the varying student needs and to support faculty teaching.

*Eight-Week Mini-Semester Courses*

To develop a competitive advantage and meet the interests of its student population, a large university in the southeastern United States piloted the 8-week mini-semester in its MBA program. The following reasons served as the rationale to adapt traditional 16-week courses into compressed 8-week mini-semesters:

1. Provide applicants with additional "entry" points into the program. Traditionally, students could begin the MBA program at the beginning of the Fall semester (August) or the Spring semester (January). Thus, a student expressing interest in enrolling after the traditional start date of a semester had to wait, in some cases for several months, before starting the program. The administration felt that some students developed "cold feet" during this waiting period and revoked their admissions, thus impacting their potential enrollment.

2. Offer students employed in seasonal industries, such as retail or accounting, the possibility of completing a course before their identified "busy" seasons. For example, students employed in retail businesses often elected to not take courses during the Fall semester, anticipating that courses may conflict with their work schedules during the holidays. Similarly, students in some accounting or finance fields would not take courses during the Spring semester because of the demands of the tax season. Having the opportunity to take a mini-semester course during the first or second mini-semester would allow students employed in either field to still take a course and not lose a semester of progress toward completing their MBA degrees.

3. Allow students to focus on one subject per 8 weeks but still complete two courses in a semester, instead of taking two 16-week course subjects concurrently. Taking one course during each mini-semester still allows students to complete two courses in a 16-week period, but without overloading the student with two different topics.

4.  Maintain competitiveness. A review of U.S. universities found that 134 institutions offered some form of mini-semester program. Thus, not having such a program could potentially steer students, particularly online students, to another university.

While the above reasons are compelling from a business and marketing standpoint for students, the faculty expressed concerns about compressing 16 weeks of course material into the 8-week sessions. The faculty concerns included:

1.  Are students able to learn the material at a fast pace?
2.  For project-oriented courses, is there enough available time to complete the project?
3.  Will the students who take both 8-week and 16-week courses concurrently become confused by having to adhere to two different schedules?
4.  Can the faculty balance the added workload when they are assigned to teach two sections of the same course, but with each on a different schedule? Some faculty members viewed this as though they had to prepare for two different courses.

To address faculty concerns, internally conducted Internet research located 134 institutions which offered abbreviated semester MBA classes. Some universities offered mini semesters of different durations. This evidence from other universities provided support to the thesis that it was important to explore the compressed course options to remain competitive with other institutions and serve as an important marketing tool to prospective students.

At our and other U.S. institutions, most MBA students are working professionals who are seeking credentials to advance in their careers. Thus, they want to balance their professional and personal lives with their academic pursuits. The decision of how many courses to complete each semester is dependent on their other responsibilities and the program design. In this study, we evaluated students who completed a course in the traditional 16-week semester versus students who completed the same course in 8 weeks. The course content was the same, but 8-week courses were completed in half the time, requiring students to presumably double their efforts. The number and type of courses students take per semester is a personal decision.

## 3. Course Redesign

Based on the above research, a pilot program to test the impact of the 8-week mini-semester on student engagement and satisfaction was launched at a large university in the southeastern United States. The Department of Marketing and Supply Chain Management volunteered to conduct the pilot program because it offered both quantitative (Supply Chain) and qualitative courses (Marketing) in both face-to-face and online modes. The quantitative courses are considered by most students to be more challenging than qualitative courses in an 8-week mini-semester format.

Faculty volunteers were invited to participate in the pilot program. In the pilot, the volunteer faculty members taught their course(s), one section as an online course and another section as a face-to-face section, during a 16-week regular semester. Some were then assigned the same courses to be taught in the same formats in a subsequent 8-week mini-semester, dependent on the graduate school's needs. This required that participating faculty redesign their normal 16-week courses to be delivered in the 8-week block. Below, an example of a course redesign for the quantitative course is explained. Similar activities were implemented in the qualitative marketing courses, including course quizzes, additional discussion boards, and more streamlined content.

### 3.1. Redesign of Face-to-Face Quantitqative Courses into 8-Week Mini-Semester Hybrid Courses

The three participating faculty members from the quantitative area redesigned their courses to the 8-week mini-semester format according to their teaching preferences. The following is a representative example of the redesign for the required MBA course in Business Analytics. During a regular 16-week semester, the course meets twice a week for 1 h and 15 min. It is taught in a flipped format.

As discussed by Swart and Wuensch, the flipped format requires both in- and out-of-class responsibilities for the student and the instructor [8]. For each class, the instructor had to:

1. Establish daily learning objectives.
2. Provide daily lectures on appropriate audio/visual media.
3. Prepare the Interactive Group Learning (IGL) activity to accomplish the daily learning objectives.
4. Prepare daily quizzes to test that everyone in each group had accomplished the learning objectives.

Prior to each class meeting, students were required to study the daily lecture and prepare questions that they wanted answered in class. During class, the following sequence of events took place:

- *Instructor*: Make announcements and comments about the day's material.
- *Students*: Ask questions (if any).
- *Instructor*: Provide daily IGL activity (usually a problem) to groups.
- *Students*: Collaborate to complete the daily IGL activity. Ask for coaching from instructor (as necessary). The work is usually shown on a large screen.
- *Instructor:* Provide coaching (as necessary). Coach/teach by walking around (observe group activities and intervene when appropriate to point out mistakes, etc.).
- *Instructor/Students*: Students and instructor agree that the objectives of the IGL activity have been achieved.
- *Students*: Take daily quiz to demonstrate that everyone has achieved that day's learning goals.

The 8-week mini-semester requires that classes meet with double the frequency to that of the 16-week regular semester, thus meeting four times a week, if the course has face-to-face delivery. This creates a classroom availability conflict since most universities schedule classes either on a Monday, Wednesday, Friday or a Tuesday, Thursday schedule. To avoid this conflict, it was decided to make the 8-week mini-semester a hybrid course. The class would meet face-to-face on Monday and Wednesday, and Tuesday and Thursday would be online. In the following section, we discuss how the online courses were redesigned for 8-week mini-semesters.

*3.2. Redesign of Online Quantitative Courses into 8-Week Mini-Semester Courses*

The 16-week online class is taught in a flipped mode, like the face-to-face class. Swart and MacLeod [9] show how that is accomplished and that there is no difference in student engagement and satisfaction nor in student willingness to recommend the course to a friend. Figures 1 and 2 show the similarities between the flipped face-to-face and online courses. The only difference in the flipped 16-week regular semester class and the 8-week mini-semester class is that instead of taking a quiz twice a week to demonstrate mastery of the material, the quizzes are given twice as frequently, requiring that students must accelerate their learning.

It also requires that the instructor be available daily to respond to student questions. Although the Q&A sessions are completed informally, the instructor also has formal twice-a-week Q&A sessions which are recorded for those whose schedules do not allow them to be present. Students are given copies of the 8-week course syllabus and schedule in advance, prior to the course starting, so that they can determine if the requirements and intensity fit their schedules. If not, students switch to a 16-week semester section of the course.

The syllabi and course schedule of the online and face-to-face sections of the courses are identical, not only for the purposes of this study but also to facilitate a transition of the face-to-face classes to online in the event of natural, manmade, or professional/personal disruptions.

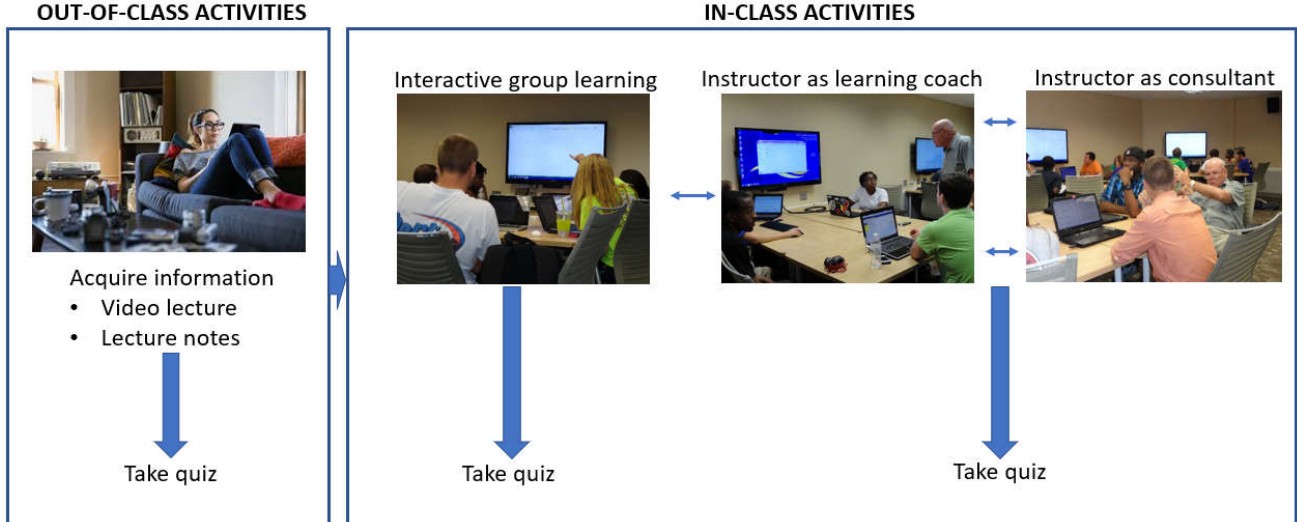

**Figure 1.** The face-to-face flipped classroom.

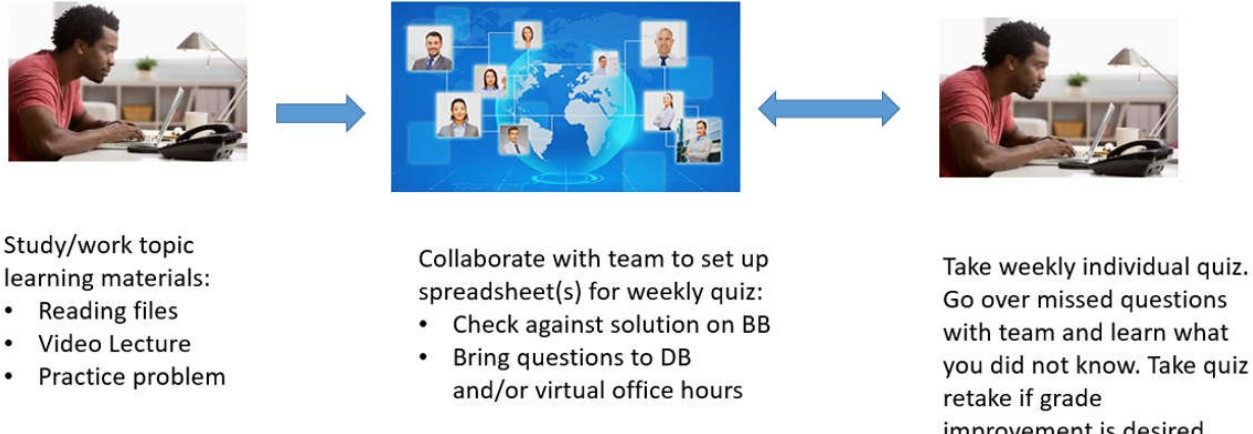

**Figure 2.** The online flipped class.

## 4. Measuring Student Engagement and Satisfaction

The theoretical foundation for our approach to measuring student engagement and satisfaction is Michael G. Moore's Theory of Transactional Distance [10]. He postulated that distance could be measured from a social science point-of-view rather than in a physical sense, and that promoting engaged learning required three kinds of interaction: learner to instructor, learner to content, and learner to learner [11].

Zhang extended Moore's work to include the interaction of the learner with the online learning environment that was beginning to emerge at that time [12]. She redefined transactional distance (TD) as four "barriers to learning": TD between student and student (TDSS), between student and teacher (TDST), between student and content (TDSC), and between student and the online learning environment (TDSI). She developed a survey instrument called the "Scale of Transactional Distance".

Swart, MacLeod, Paul, Zhang, and Gagulic developed the Relative Proximity Theory (RPT) by converting the Scale of Transactional Distance to a disconfirmation approach, recording two responses for each element, once for the actual course delivery and once for what the student considered to be an ideal course delivery [13]. They confirmed the goodness of fit and validation of the original scale as altered to a disconfirmation approach. Being able to quantitatively "measure" how close a course delivery is to "ideal" has the major benefit of indicating not only what the level of satisfaction is, but also how much room there is for improvement. This scale was updated by Paul et al. and labelled the "Scale of Relative Proximity of Transactional Distance" (SRPTD) [14].

The SRPTD is shown in Table 1. Each of the relative proximities of transactional distances are composed of several elements. For example, the relative proximity of transactional distance between student and instructor (ΔTDST) is composed of three elements: Δtdst1, Δ tdst2, and Δtdst3. To assess the relative proximity of ΔTDST, the student is given a survey that asks two questions for every element. For tdst1, these questions are shown in Figure 3. The student responds to each on a five-point Likert scale (5 = untrue; 0 = true). The difference in the numerical values of the responses constitutes a measure of how close to perfect this instructor is in providing feedback.

**Table 1.** Scale of Relative Proximity of Transactional Distance (SRPTD).

| Factor | Element | Description/Question |
|---|---|---|
| ΔTDST | | Transactional Distance between Student and Instructor |
| | Δtdst1 | I receive prompt feedback from the instructor on my academic performance |
| | Δtdst2 | The instructor was helpful to me |
| | ΔΔtdst3 | The instructor can be turned to when I need help in the course |
| ΔTDSC | | Transactional Distance between Student and Content |
| | Δtdsc1 | This course emphasized synthesizing and organizing ideas, information, or experiences into new, more complex interpretations and relationships |
| | Δtdsc2 | This course emphasized making judgements about the value of information, arguments, or methods such as examining how others gathered and incorporated data and assessing the soundness of their conclusions |
| | Δtdsc3 | This course emphasized the application of theories and concepts to practical problems or in new situations |
| ΔTDSS | | Transactional Distance between Student and Student |
| | Δtdss1 | I get along well with my classmates |
| | Δtdss2 | I feel valued by the class members in this online class |
| | Δtdss3 | My classmates in this online class value my ideas and opinions very highly |
| | Δtdss4 | My classmates respect me in this online class |
| | Δtdss5 | The class members are supportive of my ability to make my own decisions |
| ΔTDSTECH | | Transactional Distance between Student and Technology |
| | Δtdstech1 | I experienced frustration using my institution's Learning Management System |
| | Δtdstech2 | I had to consciously think about how to use my institution's Learning Management System |
| | Δtdstech3 | I find it pleasant to use my institution's Learning Management System |
| ΔSatWL | | Outcomes |
| | ΔSatWL1 | I benefit from this course |
| | ΔSatWL2 | This course met my expectations |
| | ΔSatWL3 | I experienced and learned new things in this course |
| | ΔSatWL4 | The content covered in this course was not interesting |
| | ΔSatWL5 | I would like to take more course like this one |
| | ΔSatWL6 | I wish other courses were more like this one |

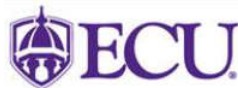

I receive prompt feedback from the instructor on my academic performance.

|  | Untrue | More untrue that true | Neither true or untrue | More true than untrue | True |
|---|---|---|---|---|---|
| ACTUAL experience in this course | ○ | ○ | ○ | ○ | ○ |
| IDEAL experience | ○ | ○ | ○ | ○ | ○ |

**Figure 3.** SRPTD survey question for Δtds1.

The average of the responses to Δtdst1, Δtdst2, and Δtdst3 is the value of ΔTDST (we assume the elements have equal weight). When this survey is given to an entire class, the average of all the student responses becomes a measure of how close to ideal this instructor was in providing prompt feedback to the class. The same process is followed for each of the transactional distances.

The relative proximities of the transactional distances can be interpreted as reflecting how far from ideal a student's engagement with the instructor is, and with the content of the course, with other students, and with the instructional technology that is being used in the course. Together, they have been shown to produce an overall outcome which is defined as student satisfaction with learning and which consists of six elements [14]. The elements are transformed to relative proximities in a similar fashion to how the transactional distances were transformed.

Using the above theoretical foundation to measure student engagement and satisfaction, we designed the online research survey, addressing concerns that students may have a hard time engaging with faster-paced courses and that their satisfaction with the course, and hence the program effectiveness, would be decreased. Using the above research questions, the online research survey was designed to address the following research propositions for both face-to-face and online students:

- P1: Students taking a mini-semester course will not have the same level of engagement as students taking the same course from the same instructor during a 16-week semester.
- P2: Students taking a mini-semester course will not have the same level of satisfaction as students taking the same course from the same instructor during a 16-week semester.
- P3: Students taking a mini-semester course will be less likely to recommend the course to a friend.

## 5. Methods

To account for possible differences in engagement and satisfaction between the quantitative and qualitative courses, two instructors teaching two different courses in the marketing (MKTG) area participated in the study and four from the quantitative area (SCM). Each course was taught in a 16-week semester during the spring of 2022 and in an 8-week mini-semester during the fall of 2022.

Table 2 lists the participants by term and gender. The noticeable difference in participants from the face-to-face and the online classes reflects our institution's MBA enrollment, which amounts to a total of 714 students, of which 96 are traditional face-to-face students. No face-to-face MKTG courses were offered in the 8-week mini semester format during the fall of 2022.

**Table 2.** Participants in the study by term and gender.

| MKTG | | Online | | | F2F | | |
| --- | --- | --- | --- | --- | --- | --- | --- |
| | | Male | Female | Total | Male | Female | Total |
| 16 Weeks | good responses | 36 | 29 | 65 | 7 | 2 | 9 |
| 8 Weeks | good responses | 35 | 29 | 64 | - | - | - |
| **OMGT** | | | | | | | |
| 16 Weeks | Good responses | 54 | 67 | 121 | 11 | 3 | 14 |
| 8 Weeks | good responses | 49 | 59 | 108 | 10 | 11 | 21 |
| | Totals | 174 | 184 | 358 | 28 | 16 | 44 | 402 |

Research proposition 1 will be addressed by having the SRPTD survey results calculated and by performing an independent samples *t*-test. Research proposition 2 will be addressed via a stepwise multiple regression with the dependent variable being satisfaction with learning (ΔSatWL) and the independent variables as shown in Table 3. Research proposition 3 will be addressed with binary logistic regression.

**Table 3.** Variables for the stepwise multiple linear regression.

| **Dependent Variable:** | |
| --- | --- |
| ΔSatWL | |
| | |
| **Independent Variables:** | |
| 16 WK SEM 1 | Indicator variable- a value of 1 indicates a 16 week semester and 0 an 8 week semester |
| MALE 1 | Indicator variable- a value of 1 indicates respondent is a self declared male and 0 a self declared female. answer |
| MKTG 1 | Indicator variable- a value of 1 indicates the course was a MKTG prefix (qualitative) and 0 an OMGT prefix (quantitative) |
| ONLINE 1 | Indicator variable- a value of 1 indicates the course was online and 0 a f2f course |
| ΔTDSC | Transactional distance between student and course content |
| ΔTDSS | Transactional distance between student and student (classmates) |
| ΔTDST | Transactional distance between student and teacher |
| ΔTDSTECH | Transactional distance between student and the instructional technology employed in the course |

## 6. Results

Below are the results based on the identified propositions.

### 6.1. Research Proposition 1

The results of the independent sample *t*-test indicated that the only statistically significant difference in the transactional distances between the regular and mini-semester courses was between student and course content (ΔTDSC), as indicated by the asterix (*). However, that does not translate to a difference in the overall student satisfaction with learning (ΔSatWL). The results of the analysis are shown in Table 4. SPSS was used to complete the data analysis.

**Table 4.** Results of the independent sample *t*-test for differences in RPTDS.

| Transactional Distance (Δ) Between: | | 8 Week | 16 Week | *t*-Test Significance | |
| --- | --- | --- | --- | --- | --- |
| Student Satisfaction with their learning | ΔSatWL | 0.86516 | 0.84722 | 0.659 | |
| Student and Course Content | ΔTDSC | 0.40201 | 0.26765 | 0.029 | * |
| Student and Student | ΔTDSS | 0.17487 | 0.25686 | 0.106 | |
| Student and Teacher | ΔTDST | 0.37018 | 0.29392 | 0.270 | |
| Student and the instructional technology | ΔTDSTECH | 0.52429 | 0.50980 | 0.875 | |
| | N | 199 | 203 | | |

Thus, we can conclude that research proposition 1 is true only for TDSC. Note that due to the small sample size of face-to-face students, we have aggregated all students.

### 6.2. Research Proposition 2

Using the variables defined in Table 3, the results of the stepwise multiple regression (SPSS 27) indicated that the unique significant predictors on student satisfaction with learning are the transactional distance between the student and (1) course content ($\Delta$TDSC), (2) instructor ($\Delta$TDSI), and (3) instructional technology utilized in the course ($\Delta$TDSTECH). These three variables explained 35.8% of the variability of the data. The detailed results are shown in Table 5. It should be reiterated that neither the length of the semester, gender, type of course, nor the mode of delivery were among the significant predictors of student satisfaction with their learning.

**Table 5.** Stepwise multiple regression results to determine predictors of $\Delta$SatWL.

| Model | R | R Square | Adjusted R Square | Std. Error of the Estimate | Change Statistics | | | | |
|-------|---|----------|-------------------|----------------------------|-------------------|---|---|---|---|
| | | | | | R Square Change | F Change | df1 | df2 | Sig. F Change |
| 1 | 0.534 [a] | 0.285 | 0.283 | 0.346 | 0.285 | 158.960 | 1 | 399 | 0.000 |
| 2 | 0.579 [b] | 0.335 | 0.332 | 0.334 | 0.050 | 30.074 | 1 | 398 | 0.000 |
| 3 | 0.599 [c] | 0.358 | 0.354 | 0.328 | 0.023 | 14.394 | 1 | 397 | 0.000 |

| | | | Coefficients [a] | | | | | |
|---|---|---|---|---|---|---|---|---|
| | | B | Std. Error | Beta | t | Sig. | Tolerance | VIF |
| 3 | (Constant) | 0.701 | 0.020 | | 35.533 | 0.000 | | |
| | $\Delta$TDSC | 0.206 | 0.036 | 0.311 | 5.736 | 0.000 | 0.551 | 1.814 |
| | $\Delta$TDSI | 0.140 | 0.033 | 0.237 | 4.208 | 0.000 | 0.510 | 1.960 |
| | $\Delta$TDSTECH | 0.077 | 0.020 | 0.173 | 3.794 | 0.000 | 0.780 | 1.282 |

### 6.3. Research Proposition 3

A binary variable (yes/no) asking students whether they would recommend the course to a friend was added to the SRPTD survey. In the sample, 309 students replied that they would recommend the course, while 90 said they would not. To determine which factors are significant predictors of willingness to recommend, a binary logistic regression was performed using the same variables as in the stepwise regression. The results, shown in Table 6, indicated that:

1. Students on the qualitative courses (MKTG) have a greater likelihood of recommending the 8-week course.
2. Online students have a lower likelihood of recommending the 8-week course.
3. As $\Delta$TDSC becomes smaller, the likelihood of recommending the 8-week course becomes larger (recall that smaller means closer to ideal and hence smaller is better).
4. As $\Delta$TDSI becomes smaller, the likelihood of recommending the 8-week course becomes larger.

**Table 6.** Results of the binary logistic regression (Nagelkerke R Square = 0.444).

| | Variables in the Equation | | | | | | |
|---|---|---|---|---|---|---|---|
| | | B | S.E. | Wald | df | Sig. | Exp(B) |
| | 16 WK SEM 1 | 0.297 | 0.306 | 0.937 | 1 | 0.333 | 1.345 |
| | MALE 1 | 0.427 | 0.31 | 1.903 | 1 | 0.168 | 1.533 |
| | MKTG 1 | 0.777 | 0.361 | 4.635 | 1 | 0.031 | 2.174 * |
| | ONLINE 1 | −1.644 | 0.807 | 4.147 | 1 | 0.042 | 0.193 * |
| Step 1 [a] | $\Delta$TDSC | −1.399 | 0.324 | 18.669 | 1 | 0 | 0.247 * |
| | $\Delta$TDSS | 0.046 | 0.266 | 0.03 | 1 | 0.862 | 1.047 |
| | $\Delta$TDST | −1.382 | 0.289 | 22.826 | 1 | 0 | 0.251 * |
| | $\Delta$TDSTECH | −0.078 | 0.171 | 0.208 | 1 | 0.649 | 0.925 |
| | Constant | 3.378 | 0.828 | 16.645 | 1 | 0 | 29.309 |

While the likelihoods starred above are statistically significant, the corresponding impact on the respective probabilities is small. For example, Table 7 showed the impact on probability of recommending the 8-week course for a change in course delivery mode for either quantitative or qualitative courses.

**Table 7.** The impact of likelihood changes on the probability of recommending.

|  | MKTG = 0 | MKTG = 1 |
|---|---|---|
| **ONLINE = 0** | 0.956854062 | 0.9796845 |
| **ONLINE = 1** | 0.810797403 | 0.9030852 |

## 7. Discussion

Our research appears to support other pedological studies on shortened semesters. Past research has found that perceived learning and overall satisfaction of graduate students at a medium-sized liberal arts institution were not significantly different between five-week and full-semester courses; however, effective communication was lacking in the intensive courses [15]. Moreover, a study of compressed face-to-face undergraduate general education courses at Brigham Young University measured workload and perceived value and found that compressed course effectiveness can be subject-specific [7]. Furthermore, a recent study evaluated undergraduate and postgraduate certificate courses (four-week and five-week programs) at Melbourne Institute of Technology and found six factors to predict the overall satisfaction of students in online intensive block mode and flipped classroom (BMFC) [16]. None of these studies were in MBA programs.

Since the MBA degree is the most popular graduate degree available, it is important to better understand student perceptions of engagement and satisfaction, as programs evolve their course delivery to meet the changing needs and demographics of students and to stay competitive in the MBA marketplace. As many universities develop compressed 8-week semesters to meet student needs, we piloted and examined the impact of compressed semesters on MBA student engagement and satisfaction in both face-to-face and online courses. While our university does not provide compressed semesters in all MBA-level courses, we were encouraged by the level of engagement and satisfaction in the compressed courses as well as the students' desire to recommend the courses to a friend, in both face-to-face and online sections.

First, when evaluating the student engagement of the traditional semester and the compressed mini-semester courses, transactional distance was measured between the student and the following: learning, course content, other students, the teacher, and the instructional technology. Students only reported a concern in the difference in the course content, but this did not translate to their overall engagement with learning, other students, and teachers. Since the mini-semester courses were shortened, the amount of content covered per week increased from the traditional semesters. To our knowledge, instructors did not change the amount of content taught between the two delivery approaches. Taking a compressed mini-semester course resulted in the same level of engagement as students taking the same course from the same instructor during a 16-week semester. Since the marketing courses were content-driven, not quantitative, the format of delivery may be easier for students to engage within the compressed semester, while the quantitative courses may require more engagement with the content and the instructor, supporting Lutes' and Davies' research finding that compressed course effectiveness may be content-specific [7].

Moreover, when evaluating student satisfaction, there was no reason to believe that students were more satisfied with 16-week semester courses than the compressed mini-semesters, as supported by the Ferguson and DeFelice study with liberal arts students [15]. The transactional distances between course content, instructor, and instructional technology utilized in the course are significant predictors of student satisfaction with both types of course delivery. There is no reason to believe that 8-week mini-semesters have or will impact student satisfaction.

Lastly, a majority of the sample (309 students) would recommend the course to a friend, while only 90 students said they would not. Results showed that students in MKTG and face-to-face courses had a greater likelihood of recommending the course. Neither the length of term nor the gender of the student had an impact on a student's likelihood of recommending the course. Since students were more willing to recommend either course delivery to friends, other MBA courses should be piloted to better understand if this finding was content-driven.

This research has a few limitations to address. First, the mini-semester courses were only offered in the Department of Marketing and Supply Chain Management. Adding additional business courses to this student model will enhance our understanding of the overall MBA student and course delivery options. Second, students self-selected to respond to the survey, which may have limited the results. While instructors encouraged students to complete the survey, not all instructors provide incentives for completion, which limited the sample size. Lastly, the study did not collect and evaluate the instructors' thoughts on student engagement and satisfaction.

More than 130 universities across the United States offer compressed MBA courses in their course schedules. To maintain a competitive edge with the changing needs of MBA students, MBA programs should consider the potential benefits, satisfaction, and learning opportunities from the offering of compressed mini-semester courses.

**Author Contributions:** Conceptualization, W.S.; Methodology, W.S.; Validation, C.M.K.; Formal analysis, W.S. and C.M.K.; Investigation, C.M.K.; Writing—original draft, W.S. and C.M.K.; Writing—review & editing, W.S. and C.M.K. All authors have read and agreed to the published version of the manuscript.

**Funding:** This research received no external funding.

**Institutional Review Board Statement:** The study was approved by the Research Integrity & Compliance (ORIC) of East Carolina University (UMCIRB 18-000457 and 3/9/2018) for studies involving humans.

**Informed Consent Statement:** Informed consent was obtained from all subjects involved in the study.

**Data Availability Statement:** The raw data supporting the conclusions of this article will be made available by the authors on request.

**Conflicts of Interest:** The authors declare no conflict of interest.

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
