# Peer review of "The Impact of Compressing MBA Courses on Student Satisfaction: Empirical Results"

_education, doi:10.3390/educsci14040388_

Round 1

Reviewer 1 Report

Comments and Suggestions for Authors

A well-written and well-edited study about a development work to create "fast courses" in eight weeks instead of sixteen. I can read that it is the same content in the courses, whether they are eight or 16 weeks long, but I don't know whether they give the same credits. What does studying full-time mean, for example? Is it to read a 16-week course per semester? Somewhere I read that students normally take two 16-week courses in parallel. I want clarification as to whether the question concerns the students studying twice as much (for example 200% instead of 100%) or whether it is, as I still interpret it, that they read a course content twice as fast, but at the same time have a course less to relate to? in the article.

I am missing previous research in the area. Even if the phenomenon is new in the USA, there are similar arrangements in several European countries. At the university where I work, we have, for example, students taking one course at a time (100%), equal to two courses (each 50% during the same period, together with students who take only one course (50% and work beside), as well as implementing "fast track", i.e. regular study-programs where the students study a program that normally takes three full-time semesters for only one year. There should thus be research to relate to that can elevate the article from an account of the development work to become part of the research field in the area.

Apart from this, I only have one comment: there are too many bullet lists for my taste. Most numbered, but one without a number.

Author Response

Thank you to the reviewers for the comments and suggestions to improve the manuscript.  We have responded to each of the reviewers’ comments below and in the manuscript.  Our responses are below in red and the updates are highlighted in red in the revised manuscript.

Reviewer 2 Report

Comments and Suggestions for Authors

Dear authors,

Thank you for sending your manuscript to the journal. You made great efforts, but there are some flaws, especially in the introduction, where you did not discuss the research problem and your study goal. Please state it clearly. 

Methodology

Please write more about your research methodology and how you develop the tool to measure student satisfaction. 

Data analysis 

Please add the procedures for data analysis

Discussion is superficial. Please discuss your findings deeply and connect it with the findings of previous studies. 

Author Response

(The authors gave the same response as above.)

Round 2

Reviewer 2 Report

Comments and Suggestions for Authors

The authors addressed all of my previous comments